# AREA ATTENTION

## ABSTRACT

Existing attention mechanisms, are mostly item-based in that a model is trained to attend to individual items in a collection (the memory) where each item has a predefined, fixed granularity, e.g., a character or a word. Intuitively, an area in the memory consisting of multiple items can be worth attending to as a whole. We propose area attention: a way to attend to an area of the memory, where each area contains a group of items that are either spatially adjacent when the memory has a 2-dimensional structure, such as images, or temporally adjacent for 1-dimensional memory, such as natural language sentences. Importantly, the size of an area, i.e., the number of items in an area or the level of aggregation, is dynamically determined via learning, which can vary depending on the learned coherence of the adjacent items. By giving the model the option to attend to an area of items, instead of only individual items, a model can attend to information with varying granularity. Area attention can work along multi-head attention for attending to multiple areas in the memory. We evaluate area attention on two tasks: neural machine translation (both character and token-level) and image captioning, and improve upon strong (state-of-the-art) baselines in all the cases. These improvements are obtainable with a basic form of area attention that is parameter free. In addition to proposing the novel concept of area attention, we contribute an efficient way for computing it by leveraging the technique of summed area tables.

## 1 INTRODUCTION

Attentional mechanisms have significantly boosted the accuracy on a variety of deep learning tasks (Bahdanau et al., 2014; Luong et al., 2015; Xu et al., 2015). They allow the model to selectively focus on specific pieces of information, which can be a word in a sentence for neural machine translation (Bahdanau et al., 2014; Luong et al., 2015) or a region of pixels in image captioning (Xu et al., 2015; Sharma et al., 2018).

An attentional mechanism typically follows a *memory-query* paradigm, where the memory $M$ contains a collection of items of information from a source modality such as the embeddings of an image or the hidden states of encoding an input sentence, and the query $q$ comes from a target modality such as the hidden state of a decoder model. In recent architectures such as Transformer (Vaswani et al., 2017), self-attention involves queries and memory from the same modality for either encoder or decoder. Each item in the memory has a key and value $(k_i, v_i)$, where the key is used to compute the probability $a_i$ regarding how well the query matches the item (see Equation 1).

$$a_i = \frac{\exp(f_{att}(q, k_i))}{\sum_{j=1}^{|M|} \exp(f_{att}(q, k_j))} \tag{1}$$

The typical choices for $f_{att}$ include dot products $qk_i$ (Luong et al., 2015) and a multilayer perceptron (Bahdanau et al., 2014). The output $O_q^M$ from querying the memory $M$ with $q$ is then calculated as the sum of all the values in the memory weighted by their probabilities (see Equation 2), which can be fed to other parts of the model for further calculation. During training, the model learns to attend to specific piece of information, e.g., the correspondance between a word in the target sentence and a word in the source sentence for translation tasks.

$$O_q^M = \sum_{i=1}^{|M|} a_i v_i \tag{2}$$

Attention mechanisms are typically designed to focus on individual items in the entire memory, where each item defines the granularity of what the model can attend to. For example, it can be a character for a character-level translation model, a word for a word-level model or a grid cell for an image-based model. Such a construction of attention granularity is predetermined rather than learned. While this kind of item-based attention has been helpful for many tasks, it can be fundamentally limited for modeling complex attention distribution that might be involved in a task.

In this paper, we propose *area attention*, as a general mechanism for the model to attend to a group of items in the memory that are structurally adjacent. In area attention, each unit for attention calculation is an area that can contain one or more than one item. Each of these areas can aggregate a varying number of items and the granularity of attention is thus learned from the data rather than predetermined. Note that area attention subsumes item-based attention because when an area contains a single item, it is equivalent to regular attention mechanisms. Area attention can be used along multi-head attention (Vaswani et al., 2017). With each head using area attention, multi-head area attention allows the model to attend to multiple areas in the memory. As we show in the experiments, the combination of both achieved the best results.

Extensive experiments with area attention indicate that area attention outperforms regular attention on a number of recent models for two popular tasks: machine translation (both token and character-level translation on WMT'14 EN-DE and EN-FR), and image captioning (trained on COCO and tested for both in-domain with COCO40 and out-of-domain captioning with Flickr 1K). These models involve several distinct architectures, such as the canonical LSTM seq2seq with attention (Luong et al., 2015) and the encoder-decoder Transformer (Vaswani et al., 2017; Sharma et al., 2018).

## 2 RELATED WORK

Item-grouping has been brought up in a number of language-specific tasks. Ranges or segments of a sentence, beyond individual tokens, have been often considered for problems such as dependency parsing or constituency parsing in natural language processing. Recent works (Wang & Chang, 2016; Stern et al., 2017; Kitaev & Klein, 2018) represent a sentence segment by subtracting the encoding of the first token from that of the last token in the segment, assuming the encoder captures contextual dependency of tokens. The popular choices of the encoder are LSTM (Wang & Chang, 2016; Stern et al., 2017) or Transformer (Kitaev & Klein, 2018). In contrast, the representation of an area (or a segment) in area attention, for its basic form, is defined as the mean of all the vectors in the segment where each vector does not need to carry contextual dependency. We calculate the mean of each area of vectors using subtraction operation over a summed area table (Viola & Jones, 2001) that is fundamentally different from the subtraction applied in these previous works.

Lee et al. proposed a rich representation for a segment in coreference resolution tasks (Lee et al., 2017), where each span (segment) in a document is represented as a concatenation of the encodings of the first and last words in the span, the size of the span and an attention-weighted sum of the word embeddings within the span. Again, this approach operates on encodings that have already captured contextual dependency between tokens, while area attention we propose does not require each item to carry contextual or dependency information. In addition, the concept of range, segment or span that is proposed in all the above works addresses their specific task, rather than aiming for improving general attentional mechanisms.

Previous work have proposed several methods for capturing structures in attention calculation. For example, Kim et al. used a conditional random field to directly model the dependency between items, which allows multiple "cliques" of items to be attended to at the same time (Kim et al., 2017). Niculae and Blondel approached the problem, from a different angle, by using regularizers to encourage attention to be placed onto contiguous segments (Niculae & Blondel, 2017). In image captioning tasks, Pedersoli et al. enabled a model to attend to object proposals on an image (Pedersoli et al., 2016) while You et al. applied attention to semantic concepts and visual attributes that are extracted from an image (You et al., 2016).

Compared to these previous works, area attention we propose here does not require to train a special network or sub-network, or use an additional loss (regularizer) to capture structures. It allows a model to attend to information at a varying granularity, which can be at the input layer where each item might lack contextual information, or in the latent space. It is easy to apply area attention to

existing single or multi-head attention mechanisms. By enhancing Transformer, an attention-based architecture, (Vaswani et al., 2017) with area attention, we achieved state-of-art results on a number of tasks.

## 3 AREA-BASED ATTENTION MECHANISMS

An area is a group of structurally adjacent items in the memory. When the memory consists of a sequence of items, a 1-dimensional structure, an area is a range of items that are sequentially (or temporally) adjacent and the number of items in the area can be one or multiple. Many language-related tasks are categorized in the 1-dimensional case, e.g., machine translation or sequence prediction tasks. In Figure 1, the original memory is a 4-item sequence. By combining the adjacent items in the sequence, we form area memory where each item is a combination of multiple adjacent items in the original memory. We can limit the maximum area size to consider for a task. In Figure 1, the maximum area size is 3.

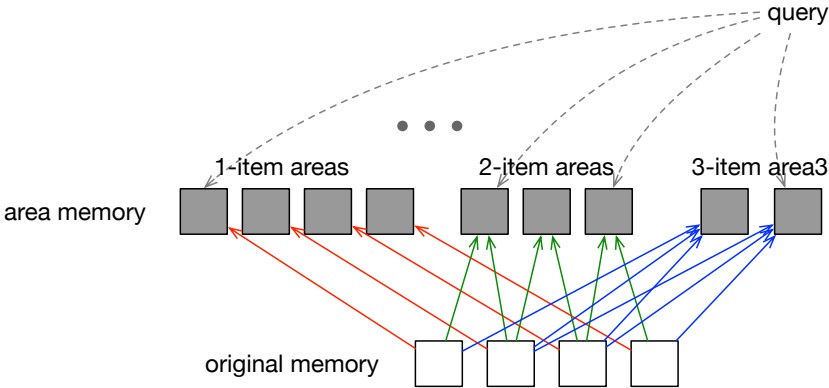

Figure 1: An illustration of area attention for the 1-dimensional case. In this example, the memory is a 4-item sequence and the maximum size of an area allowed is 3.

When the memory contains a grid of items, a 2-dimensional structure, an area can be any rectangular region in the grid (see Figure 2). This resembles many image-related tasks, e.g., image captioning. Again, we can limit the maximum size allowed for an area. For a 2-dimensional area, we can set the maximum height and width for each area. In this example, the original memory is a 3x3 grid of items and the maximum height and width allowed for each area is 2.

As we can see, many areas can be generated by combining adjacent items. For the 1-dimensional case, the number of areas that can be generated is $|R| = (L - S)S + (S + 1)S/2$ where $S$ is the maximum size of an area and $L$ is the length of the sequence. For the 2-dimensional case, there are an quadratic number of areas can be generated from the original memory: $|R| = |R_v||R_h|$. $|R_v| = (L_v - H)H + (H + 1)H/2$ and $|R_h| = (L_h - W)W + (W + 1)W/2$ where $L_v$ and $L_h$ are the height and width of the memory grid and $H$ and $W$ are the maximum height and width allowed for a rectangular area.

To be able to attend to each area, we need to define the key and value for each area that contains one or multiple items in the original memory. As the first step to explore area attention, we define the key of an area, $\mu_i$, simply as the mean vector of the key of each item in the area.

$$\mu_i = \frac{1}{|r_i|} \sum_{j=1}^{|r_i|} k_{i,j} \tag{3}$$

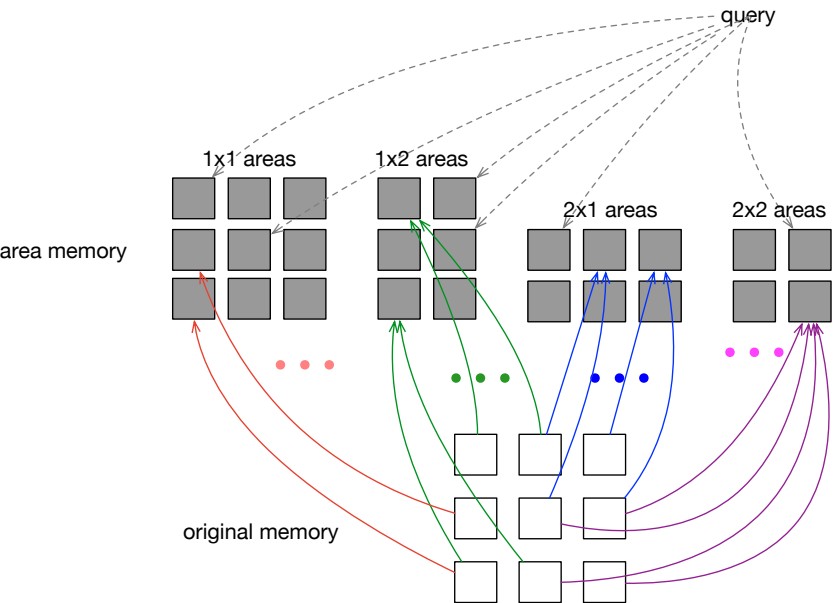

Figure 2: An illustration of area attention for the 2-dimensional case. In this example, the memory is a 3x3 grid and the dimension allowed for an area is 2x2.

where $|r_i|$ is the size of the area $r_i$. For the value of an area, we simply define it as the the sum of all the value vectors in the area.

$$v_i^{r_i} = \sum_{j=1}^{|r_i|} v_{i,j} \tag{4}$$

With the keys and values defined, we can use the standard way for calculating attention as discussed in Equation 1 and Equation 2. Note that this basic form of area attention (Eq.3 and Eq.4) is parameter-free—it does not introduce any parameters to be learned.

## 3.1 COMBINING AREA FEATURES

Alternatively, we can derive a richer representation of each area by using features other than the mean of the key vectors of the area. For example, we can consider the standard deviation of the key vectors within each area.

$$\sigma_i = \sqrt{\frac{1}{|r_i|} \sum_{l=1}^{|r_i|} (k_{i,l} - \mu_i)^2} \tag{5}$$

We can also consider the height and width of each area, $h_i, 1 \leq h_i \leq H$ and $w_i, 1 \leq w_i \leq W$, as the features of the area. To combine these features, we use a multi-layer perceptron. To do so, we treat $h_i$ and $w_i$ as discrete values and project them onto a vector space using embedding (see Equation 6 and 7).

$$e_i^h = 1(h_i)E^h \tag{6}$$

$$e_i^w = 1(w_i)E^w \tag{7}$$

where $1(h_i)$ and $1(w_i)$ are the one-hot encoding of $h_i$ and $w_i$, and $E^h \in R^{H \times S}$ and $E^w \in R^{W \times S}$ are the embedding matrices. $S$ is the depth of the embedding. We concatenate them to form the representation of the shape of an area.

$$e_i = [e_i^h, e_i^w] \tag{8}$$

We then combine them using a single-layer perceptron followed by a linear transformation (see Equation 9).

$$k_i^{\tau} = \phi(\mu_i W_\mu + \sigma_i W_\sigma + e_i W_e) W_d \tag{9}$$

where $\phi$ is a nonlinear transformation such as ReLU, and $W_\mu \in \mathbb{R}^{D \times D}$, $W_\sigma \in \mathbb{R}^{D \times D}$, $W_e \in \mathbb{R}^{2S \times D}$ and $W_d \in \mathbb{R}^{D \times D}$. $W_\mu$, $W_\sigma$, $W_e$ and $W_d$ are trainable parameters.

## 3.2 Fast Computation Using Summed Area Table

If we naively compute $\mu_i$, $\sigma_i$ and $v_i^{\tau_i}$, the time complexity for computing attention will be $O(|M|A^2)$ where $|M|$ is the size of the memory that is $L$ for a 1-dimensional sequence or $L_v L_h$ for a 2-dimensional memory. $A$ is the maximum size of an area, which is $S$ in the one dimensional case and $WH$ in the 2-dimensional case. This is computationally expensive in comparison to the attention computed on the original memory, which is $O(|M|)$. To address the issue, we use summed area table, an optimization technique that has been used in computer vision for computing features on image areas (Viola & Jones, 2001). It allows constant time to calculate a summation-based feature in each rectangular area, which allows us to bring down the time complexity to $O(|M|A)$—We will report on the actual time cost in our experimental section.

Summed area table is based on a pre-computed integral image, $I$, which can be computed in a single pass of the memory (see Equation 10). Here let us focus on the area value calculation for a 2-dimensional memory because a 1-dimensional memory is just a special case with the height of the memory grid as 1.

$$I_{x,y} = v_{x,y} + I_{x,y-1} + I_{x-1,y} \tag{10}$$

where $x$ and $y$ are the coordinates of the item in the memory. With the integral image, we can calculate the key and value of each area in constant time. The sum of all the vectors in a rectangular area can be easily computed as the following (Equation 11).

$$v_{x1,y1,x2,y2} = I_{x2,y2} + I_{x1,y1} - I_{x2,y1} - I_{x1,y2} \tag{11}$$

where $v_{x1,y1,x2,y2}$ is the value for the area located with the top-left corner at $(x_1, y_1)$ and the bottom-right corner at $(x_2, y_2)$. By dividing $v_{x1,y1,x2,y2}$ with the size of the area, we can easily compute $\mu_{x1,y1,x2,y2}$. Based on the summed area table, $\sigma_{x1,y1,x2,y2}^2$ (thus $\sigma_{x1,y1,x2,y2}$) can also be computed at constant time for each area (see Equation 12), where $I_{x,y}^2 = v_{x,y}^2 + I_{x,y-1}^2 + I_{x-1,y}^2$, which is the integral image of the element-wise squared memory.

$$\sigma_{x1,y1,x2,y2}^2 = \frac{I_{x2,y2}^2 + I_{x1,y1}^2 - I_{x2,y1}^2 - I_{x1,y2}^2}{(x2 - x1) \times (y2 - y1)} - \mu_{x1,y1,x2,y2}^2 \tag{12}$$

The core component for computing these quantities is to be able to quickly compute the sum of vectors in each area after we obtain the integral image table $I$ for each coordinate $[x, y]$, as shown in Equation 10 and 11. We present the Pseudo code for performing Equation 10 and Equation 11 as well as the shape size of each area in Algorithm (1) and the code for computing the mean, sum and standard deviation (Equation 12) in Algorithm (2). These Pseudo code are designed based on Tensor operations, which can be implemented efficiently using libraries such as TensorFlow [1] and PyTorch [2].

---

[1]https://github.com/tensorflow/tensorflow
[2]https://github.com/pytorch/pytorch

---

**Algorithm 1:** Compute the vector sum and the size of each area, for all the qualified rectangular areas on a given grid.

---

**Input:** A tensor $G$ in shape of $[H, W, D]$ that represents a grid with height $H$ and width $W$ where each item is a vector of depth $D$.

**Output:** Sum of vectors of each area, $U$, and height and width of each area, $S_h$ and $S_w$.

**Hyperparameter**: maximum area width $W_a$ and height $H_a$ allowed.

1 Compute horizontal integral image $I_h$ by cumulative sum along horizontal direction over $G$;

2 Compute integral image $I_{hv}$ by cumulative sum along vertical directions over $I_h$;

3 Acquire $I$ by padding all-zero vectors to the left and top of $I_{hv}$;

4 **for** $h = 1, \cdots, H_a$ **do**

5     **for** $w = 1, \cdots, W_a$ **do**

6         $I_1 \leftarrow I[h + 1 :, w + 1 :, :]$ ;

7         $I_2 \leftarrow I[: -h - 1, : -w - 1, :]$ ;

8         $I_3 \leftarrow I[h + 1 :, : -w - 1, :]$ ;

9         $I_4 \leftarrow I[: -h - 1, w + 1 :, :]$ ;

10         $\bar{U} = I_1 + I_2 - I_3 - I_4$ ;

11         $\bar{S_h} \leftarrow [h]^{(H-h) \times (W-w)}$; Fill tensor with value $h$ for the height of each area;

12         $\bar{S_w} \leftarrow [w]^{(H-h) \times (W-w)}$; Fill tensor with value $w$ for the width of each area;

13         $S_h \leftarrow [S_h \ \bar{S_h}]$, reshape $\bar{S_h}$ to $[-1, 1]$ and concatenate on the first dimension;

14         $S_w \leftarrow [S_w \ \bar{S_w}]$, reshape $\bar{S_w}$ to $[-1, 1]$ and concatenate on the first dimension;

15         $U \leftarrow [U \ \bar{U}]$, reshape $\bar{U}$ to $[-1, D]$ and concatenate on the first dimension;

16 **return** $U$, $S_h$ and $S_w$.

---

---

**Algorithm 2:** Compute the vector mean, standard deviation, and sum as well as the size of each area, for all the qualified rectangular areas on a grid.

---

**Input:** A tensor $G$ in shape of $[H, W, D]$ that represents a grid with height $H$ and width $W$ where each item is a vector of depth $D$.

**Output:** Vector mean $\mu$, standard deviation $\sigma$ and sum $U$ as well as height $S_h$ and width $S_w$ of each area.

1 Acquire $U$, $S_h$ and $S_w$ using Algorithm 1 with input $G$;

2 Acquire $U'$ using Algorithm 1 with input $G \odot G$ where $\odot$ is for element-wise multiplication;

3 $\mu \leftarrow U \oslash S$ where $\oslash$ is for element-wise division;

4 $\mu' \leftarrow U' \oslash S$ ;

5 $\sigma \leftarrow \sqrt{\mu' - \mu \odot \mu}$ ;

6 **return** $\mu, \sigma, U$, as well as $S_h$ and $S_w$.

---

## 4 EXPERIMENTS

We experimented with area attention on two important tasks: neural machine translation (including both token and character-level translation) and image captioning, where attention mechanisms have been a common component in model architectures for these tasks. The architectures we investigate involves several popular encoder and decoder choices, such as LSTM (Hochreiter & Schmidhuber, 1997) and Transformer (Vaswani et al., 2017). The attention mechansims in these tasks include both self attention and encoder-decoder attention.

### 4.1 TOKEN-LEVEL NEURAL MACHINE TRANSLATION

Transformer has recently (Vaswani et al., 2017) established the state of art performance on WMT 2014 English-to-German and English-to-French tasks, while LSTM with encoder-decoder attention has been a popular choice for neural machine translation tasks. We use the same dataset as the one used in (Vaswani et al., 2017) in which the WMT 2014 English-German dataset contains about 4.5 million English-German sentence pairs, and the English-French dataset has about 36 million

English-French sentence pairs (Wu et al., 2016). A token is either a byte pair (Britz et al., 2017) or a word piece (Wu et al., 2016) as in the original Transformer experiments.

### 4.1.1 TRANSFORMER TOKEN-LEVEL MT EXPERIMENTS

Transformer heavily uses attentional mechanisms, including both self-attention in the encoder and the decoder, and attention from the decoder to the encoder. We vary the configuration of Transformer to investigate how area attention impacts the model. In particular, we investigated the following variations of Transformer: *Tiny* (#hidden layers=2, hidden size=128, filter size=512, #attention heads=4), *Small* (#hidden layers=2, hidden size=256, filter size=1024, #attention heads=4), *Base* (#hidden layers=6, hidden size=512, filter size=2048, #attention heads=8) and *Big* (#hidden layers=6, hidden size=1024, filter size=4096, #attention heads=16).

During training, sentence pairs were batched together based on their approximate sequence lengths. All the model variations except *Big* uses a training batch contained a set of sentence pairs that amount to approximately 32,000 source and target tokens and were trained on one machine with 8 NVIDIA P100 GPUs for a total of 250,000 steps. Given the batch size, each training step for the Transformer Base model, on 8 NVIDIA P100 GPUs, took 0.4 seconds for Regular Attention, 0.5 seconds for the basic form of Area Attention (Eq.3 and Eq.4), 0.8 seconds for Area Attention using multiple features (Eq.9 and Eq.4).

For *Big*, due to the memory constraint, we had to use a smaller batch size that amounts to roughly 16,000 source and target tokens and trained the model for 600,000 steps. Each training step took 0.5 seconds for Regular Attention, 0.6 seconds for the basic form of Area Attention (Eq.3 and 4), 1.0 seconds for Area Attention using multiple features (Eq.9 and 4). Similar to previous work, we used the Adam optimizer with a varying learning rate over the course of training—see (Vaswani et al., 2017) for details.

Table 1: The BLEU scores on token-level translation tasks for the variations of the Transformer-based architecture.

| Model | Regular Attention | | Area Attention (Eq.3 and 4) | | Area Attention (Eq.9 and 4) | |
|-------|-------|-------|-------|-------|-------|-------|
| | EN-DE | EN-FR | EN-DE | EN-FR | EN-DE | EN-FR |
| Tiny | 18.60 | 27.07 | 19.30 | 27.39 | **19.45** | **27.79** |
| Small | 22.80 | 31.91 | 22.86 | 32.47 | **23.19** | **32.97** |
| Base | 27.96 | 39.10 | 28.17 | 39.22 | **28.52** | **39.28** |
| Big | 29.43 | 40.88 | 29.48 | **41.06** | **29.68** | 41.03 |

We applied area attention to each of the Transformer variation, with the maximum area size of 5 to both encoder and decoder self-attention, and the encoder-decoder attention in the first two layers. We found area attention consistently improved Transformer on all the model variations (see Table 1), even with the basic form of area attention where no additional parameters are used (Eq.3 and Eq.4). For Transformer Base, area attention achieved BLEU scores (EN-DE: 28.52 and EN-FR: 39.28) that surpassed the previous results for both EN-DE and EN-FR. In particular, Transformer Base with area attention even outperformed the result of Transformer Big with regular attention reported previously (Vaswani et al., 2017) on EN-DE.

For EN-FR, the performance of Transformer Big with regular attention—a baseline—does not match what was reported in the Transformer paper (Vaswani et al., 2017), largely due to a different batch size and the different number of training steps used, although area attention still outperformed the baseline consistently. On the other hand, area attention with Transformer Big achieved BLEU *29.68* on EN-DE that improved upon the state-of-art result of 28.4 reported in (Vaswani et al., 2017) with a significant margin.

### 4.1.2 LSTM TOKEN-LEVEL MT EXPERIMENTS

We used a 2-layer LSTM for both encoder and decoder. The encoder-decoder attention is based on multiplicative attention where the alignment of a query and a memory key is computed as their dot product (Luong et al., 2015). We vary the size of LSTM and the number of attention heads to

investigate how area attention can improve LSTM with varying capacity on translation tasks. The purpose is to observe the impact of area attention on each LSTM configuration, rather than for a comparison with Transformer.

Because LSTM requires sequential computation along a sequence, it trains rather slow compared to Transformer. To improve GPU utilization we increased data parallelism by using a much larger batch size than training Transformer. We trained each LSTM model on one machine with 8 NVIDIA P100. For a model has 256 or 512 LSTM cells, we trained it for 50,000 steps using a batch size that amounts to approximately 160,000 source and target tokens. When the number of cells is 1024, we had to use a smaller batch size with roughly 128,000 tokens, due to the memory constraint, and trained the model for 625,000 steps.

In these experiments, we used the maximum area size of 2 and the attention is computed from the output of the decoder's top layer to that of the encoder. Similar to what we observed with Transformer, area attention consistently improves LSTM architectures in all the conditions (see Table 2).

Table 2: The BLEU scores on token-level translation tasks for the LSTM-based architecture with varying model capacities.

| #Cells | #Heads | Regular Attention | | Area Attention (Eq.3,4) | | Area Attention (Eq.9,4) | |
|---|---|---|---|---|---|---|---|
| | | EN-DE | EN-FR | EN-DE | EN-FR | EN-DE | EN-FR |
| 256 | 1 | 19.05 | 28.51 | 19.08 | 28.39 | **19.60** | **28.61** |
| 256 | 4 | 19.92 | 29 | 20.21 | 29.45 | **20.64** | **30.39** |
| 512 | 1 | 22.13 | 31.95 | **22.14** | **32.08** | 22.02 | 31.73 |
| 512 | 4 | 22.78 | 33.2 | 22.73 | 33.05 | **23.18** | **33.44** |
| 1024 | 1 | 23.8 | 31.66 | **24** | 34.57 | 23.39 | **34.70** |
| 1024 | 4 | 20.06 | 32.82 | 24.48 | 35.54 | **24.95** | **36.02** |

## 4.2 CHARACTER-LEVEL NEURAL MACHINE TRANSLATION

Compared to token-level translation, character-level translation requires the model to address significantly longer sequences, which are a more difficult task and often less studied. We speculate that the ability to combine adjacent characters, as enabled by area attention, is likely useful to improve a regular attentional mechanisms. Likewise, we experimented with both Transformer and LSTM-based architectures for this task. We here used the same dataset, and the batching and training strategies as the ones used in the token-level translation experiments.

Transformer has not been used for character-level translation tasks. We found area attention consistently improved Transformer across all the model configurations. The best result we found in the literature is $BLEU = 22.62$ reported by (Wu et al., 2016). We achieved $BLEU = 26.65$ for the English-to-German character-level translation task and $BLEU = 34.81$ on the English-to-French character-level translation task. Note that these accuracy gains are based on the basic form of area attention (see Eq.3 and Eq.4), which does not add any additional trainable parameters to the model.

Similarly, we tested LSTM architectures on the character-level translation tasks. We found area attention outperformed the baselines in most conditions (see Table 4). The improvement seems more substantial when a model is relatively small.

## 4.3 IMAGE CAPTIONING

Image captioning is the task to generate natural language description of an image that reflects the visual content of an image. This task has been addressed previously using a deep architecture that features an image encoder and a language decoder (Xu et al., 2015; Sharma et al., 2018). The image encoder typically employs a convolutional net such as ResNet (He et al., 2015) to embed the images and then uses a recurrent net such as LSTM or Transformer (Sharma et al., 2018) to encode the image based on these embeddings. For the decoder, either LSTM (Xu et al., 2015) or Transformer (Sharma et al., 2018) has been used for generating natural language descriptions. In many of these designs,

Table 3: The BLEU scores on character-level translation tasks for the Transformer-based architecture with varying model capacities.

| Model | Regular Attention | | Area Attention (Eq.3 and 4) | |
|---|---|---|---|---|
| | EN-DE | EN-FR | EN-DE | EN-FR |
| Tiny | 6.97 | 9.47 | **7.39** | **11.79** |
| Small | 12.18 | 18.75 | **13.44** | **21.24** |
| Base | 24.65 | 32.80 | **25.03** | **33.69** |
| Big | 25.24 | 33.82 | **26.65** | **34.81** |

Table 4: The BLEU scores on character-level translation tasks for the LSTM-based architecture with varying model capacities.

| #Cells | #Heads | Regular Attention | | Area Attention (Eq.3 and 4) | |
|---|---|---|---|---|---|
| | | EN-DE | EN-FR | EN-DE | EN-FR |
| 256 | 1 | 9.96 | 16.5 | **11.17** | **17.14** |
| 256 | 4 | 11.43 | 17.86 | **12.48** | **19.06** |
| 512 | 1 | 16.31 | 23.51 | **16.55** | **24.05** |
| 512 | 4 | **17.5** | **25.01** | 17.28 | 24.97 |
| 1024 | 1 | 20.45 | 28.7 | **21.17** | **28.99** |
| 1024 | 4 | 21.28 | 30.11 | **21.53** | **30.28** |

attention mechanisms have been an important component that allows the decoder to selectively focus on a specific part of the image at each step of decoding, which often leads to better captioning quality.

In this experiment, we follow a champion condition in the experimental setup of (Sharma et al., 2018) that achieved state-of-the-art results. It uses a pre-trained Inception-ResNet to generate $8 \times 8$ image embeddings, a 6-layer Transformer for image encoding and a 6-layer Transformer for decoding. The dimension of Transformer is 512 and the number of heads is 8. We intend to investigate how area attention improves the captioning accuracy, particularly regarding self-attention and encoder-decoder attention computed off the image, which resembles a 2-dimensional case for using area attention. We also vary the maximum area size allowed to examine the impact.

Similar to (Sharma et al., 2018), we trained each model based on the training & development sets provided by the COCO dataset (Lin et al., 2014), which as 82K images for training and 40K for validation. Each of these images have at least 5 groudtruth captions. The training was conducted on a distributed learning infrastructure (Dean et al., 2012) with 10 GPU cores where updates are applied asynchronously across multiple replicas. We then tested each model on the COCO40 (Lin et al., 2014) and the Flickr 1K (Young et al., 2014) test sets. Flickr 1K is out-of-domain for the trained model. For each experiment, we report CIDEr (Vedantam et al., 2014) and ROUGE-L (Lin & Och, 2004) metrics. For both metrics, higher number means better captioning accuracy—the closer distances between the predicted and the groundtruth captions. Similar to the previous work (Sharma et al., 2018), we report the numerical values returned by the COCO online evaluation server[3] for the COCO C40 test set (see Table 5),

In the benchmark model, a regular multi-head attention is used. We then experimented with several variations by adding area attention with different maximum area sizes to the first 2 layers of the image encoder self-attention and encoder-decoder (caption-to-image) attention, which both are a 2-dimensional area attention case. $2 \times 2$ stands for the maximum area size 2 by 2 and $3 \times 3$ for 3 by 3. For the $2 \times 2$ case, an area can be 1 by 1, 2 by 1, 1 by 2, and 2 by 2 as illustrated in Figure 2. $3 \times 3$ allows more area shapes.

We found models with area attention outperformed the benchmark on both CIDEr and ROUGE-L metrics with a large margin. The models with $2 \times 2$ Eq.3 and $3 \times 3$ Eq.3 are parameter free—they

---

[3]http://mscoco.org/dataset/captions-eval

Table 5: Test accuracy of image captioning models on COCO40 (in-domain) and Flickr 1K (out-of-domain) tasks.

| Model | COCO40 | | Flickr 1K | |
|---|---|---|---|---|
| | CIDEr | ROUGE-L | CIDEr | ROUGE-L |
| Benchmark (Sharma et al., 2018) | 1.032 | 0.700 | 0.359 | 0.416 |
| Benchmark Replicate | 1.034 | 0.701 | 0.355 | 0.409 |
| $2 \times 2$ Eq.3 & 4 | **1.060** | 0.704 | 0.364 | **0.420** |
| $3 \times 3$ Eq.3 & 4 | **1.060** | 0.706 | **0.377** | 0.419 |
| $3 \times 3$ Eq.9 & 4 | 1.045 | **0.707** | 0.372 | **0.420** |

do not use any additional parameters beyond the benchmark model. $3 \times 3$ achieved the best results overall. $3 \times 3$ Eq. 9 adds a small fraction of the number of parameters to the benchmark model and did not seem to improve on the parameter-free version of area attention, although it still outperformed the benchmark.

## 5 CONCLUSIONS

In this paper, we present a novel attentional mechanism by allowing the model to attend to areas as a whole. An area contains one or a group of items in the memory to be attended. The items in the area are either spatially adjacent when the memory has 2-dimensional structure, such as images, or temporally adjacent for 1-dimensional memory, such as natural language sentences. Importantly, the size of an area, i.e., the number of items in an area or the level of aggregation, can vary depending on the learned coherence of the adjacent items, which gives the model the ability to attend to information at varying granularity. Area attention contrasts with the existing attentional mechanisms that are item-based. We evaluated area attention on two tasks: neural machine translation and image captioning, based on model architectures such as Transformer and LSTM. On both tasks, we obtained new state-of-the-art results using area attention.

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
