# OpenReview forum: "Area Attention"
_ICLR.cc/2019/Conference_

### Official Review · AnonReviewer2 · 2018-10-26
**A few concerns**

**Rating:** 5
**Confidence:** 4

**Review:**

I prefer the idea of using some statistics (such as variances) of multiple items for attention.
This direction may lead to better attention units for future works.

I do not fully understand the argument, "Attention mechanisms are designed to focus on a single item in the entire memory".
In my understanding, the attention formulation has no mathematical bias to focus on a single item.
I have been working on the enterprise NMT for years, and observed many cases where the attention weights concentrate in a few (not a single), tokens.
Do you have any comments?

Could you show some concise examples that we really need to attend multiple (adjacent) items to boost the performance?
For example, in char-based machine translation case, we can mimic the area attention with the wordpiece + token-wise NMT.
For the image case, the adjacent area looks like a "super pixel".

It is unfortunate to observe that the gains on BLEU and perplexity are limited.
Since the authors do not provide any statistical tests, or a confidence interval of the scores,
I cannot be sure these gains are truly significant.
From my experiences +1.0 BLEU score is often insignificant in NMT experiments (BLEU variance is high in general).

Summary
+ A new variant of attention, allowing attention to asses statistics of multiple items (such as variances) is interesting
- Claims are not so much convincing for the need of attending multiple adjacent items.
- Gains in experiments are limited.

---

> ### Author Response · Authors · 2018-11-26
> **Responses to Reviewer2's comments**
>
> Thank you very much for your feedback. We have revised the paper substantially and here address each of your concerns.
>
> Re: “Attention mechanisms are designed to focus on a single item”
> This is indeed miscommunication in our original submission. What we intend to convey is that regular attention mechanisms are designed to focus on individual items, i.e., the granularity of attention is each item. Such granularity is predetermined and fixed, e.g., a character or a word-piece. In contrast, area attention allows a model to attend to information with varying granularity by dynamically grouping adjacent items. For example, it can be a group of adjacent regions on an image that forms an object or a "super pixel", or multiple word pieces that form a phrase. The difference with regular attention is that we do not have to decide what the proper unit is. Rather, we let the model pick on the right level of aggregation of items or raw features. Such granularity or aggregation is acquired through learning. We have clarified this point throughout the paper.
>
> # the gains on BLEU and perplexity are limited.
> We reported BLEU scores on all the translation tasks including those previously with perplexity. We also added COCO40 Official test results for the image captioning tasks. Overall, for translation tasks, while the margin is not as large as we hoped, the accuracy gain with area attention is quite consistent throughout most conditions. Particularly on Token-level EN-DE translation, area attention achieved BLEU 29.68 that improved upon the state of the art results with a big margin. For image captioning, the accuracy gain is significant for both in-domain and out-of-domain tests.
>
> We have improved the paper in many ways. Please see the complete list of changes we made in the Summary of Changes and the revised paper for details.

---

> > ### Comment · AnonReviewer2 · 2018-12-17
> > **thank you for revised paper**
> >
> > I thank the authors for their efforts in revising paper.
> >
> > I think I understand the philosophy behind the area attention.  It is possibly a right direction for development of attention tech.
> > As the other reviewers pointed out, experimental results are remain weak to fully support the necessity of the proposed framework.
> > Thus I do not change the review score.

---

> > > ### Author Response · Authors · 2018-12-17
> > > **Thanks for your further comments**
> > >
> > > While accuracy gains are not always significant, the improvements from area attention are pretty consistently seen across most conditions. While the baseline models were well tuned by previous work, we didn’t particularly tune each model to better work with area attention. We believe some hyperparameter tuning would result in better accuracy. For example, we acquired some new results (29.74 for En-De and 41.5 for En-Fr) recently by just tuning on the maximum area size. Tuning other hyperparameters such as attention dropout ratio can help realize area attention’s full potential.

---

### Official Review · AnonReviewer3 · 2018-11-01
**Experiments are not convincing**

**Rating:** 5
**Confidence:** 5

**Review:**

[Summary]
Paper “AREA ATTENTION” extends the current attention models from word level to “area level”, i.e., the combination of adjacent words. Specifically, every $r_i$ adjacent words are first merged into a new item; next a key and the value for this item is calculated based on Eqn.(3 or 7) and Eqn. (4), and then the conventional attention models are applied to these new items. The authors work on (char level) NMT and image captioning to verify the algorithm.

[Details]
1.	In the abstract, “… Using an area of items, instead of a single, we hope attention mechanisms can better capture the nature of the task …”, can you provide an example to show why “an area of items” can “better capture the nature of the task”? In particular, you need to show why the conventional attention mechanism fails.
2.	In this new proposed framework, how should we define the query for each area including multiple items like words? For example, in Figure 1, what is the query for $n$-item areas where $n=1,2,3$.
3.	Two different kinds of keys are proposed in Eqn. (3) and Eqn. (7). Any comparison between them?
4.	I am not convinced by the experimental results.
(4a) On WMT’14 En-to-Fr and En-to-De, we know that “transformer_big” can achieve better results than the three settings shown in Table 1 & 2. The results of using transformer_big are not reported. Besides, it is not necessary to use the “tiny” setting for En-to-{De, Fr} translation considering the data size.
(4b) It is widely adopted to use token-level neural machine translation. It is not convincing to work on char-level NMT only. Also, please provide the results using transformer_big setting.
(4c) There are no BLEU scores for the LSTM setting. Note that comment (4b) and (4c) are also pointed by anonymous readers.
(4d) It is really strange for me to “trained on COCO and tested on Flickr” (See the title of Table 4). It is not a common practice in image captioning literature. Even if in (Soricut et al., 2018), the authors report the results of training of COCO and test on COCO (the Table 5). Therefore, the results are not convincing. You should train on COCO and test on COCO too.
e.	What if we use different area size? I do not find the study in this paper.

[Pros & Cons]
(+) A new attempt of the attention model that tries to build the attention beyond unigrams.
(-) Experiments are not convincing.
(-) The motivation is not strong.

---

> ### Author Response · Authors · 2018-11-27
> **Responses to Reviewer3's comments**
>
> Thank you for your detailed comments. We have carefully addressed all your questions in the revision. Please see the Summary of Changes and the revised paper for details. We here respond to each of your questions.
>
> Re: "1. can you provide an example to show why “an area of items” can “better capture the nature of the task”? In particular, you need to show why the conventional attention mechanism fails."
>
> We miscommunicated our motivation in the original paper. Our motivation is to allow a model to attend to information with varying granularity. Conventional attention mechanisms attend to items with a predefined and fixed granularity, e.g., a word piece of a character in translation or a cell in a fixed grid for image captioning. With area attention, the unit of attention can be a combination of a group of adjacent items. For example, multiple cells in an image grid can make a car that can be more efficient for the word “car” in the caption to attend to. We have clarified our motivation throughout the paper.
>
> Re: "2. In this new proposed framework, how should we define the query for each area including multiple items like words? For example, in Figure 1, what is the query for $n$-item areas where $n=1,2,3$."
>
> There is no special requirement on queries. A query can be the same as in a regular attention. For example, a word in a target sentence can be a translation of a multi-word phrase (an area) in a source sentence.
>
> Re: "3. Two different kinds of keys are proposed in Eqn. (3) and Eqn. (7). Any comparison between them?"
>
> We added a comparison of these two forms of keys in Token-Level Translation Tasks (see Table 1 & 2). Overall, keys with feature combination (Eqn.5-9) generally performs better than the basic form Eqn.3, although the basic form already outperforms the baselines in most conditions.
>
> Re: "(4a) (4b)"
>
> We added Token-Level Translation tasks in the revised paper (see Section 4.1). We also added a comparison with “Transformer_Big”. Again, area attention consistently improves these models over all the conditions. In particular, area attention achieved BLEU 29.68 on EN-DE that improved upon the previous result of BLEU 28.4 reported for Transformer Big in the original Transformer paper. For EN-FR, area attention also outperformed the baseline, although it didn’t match the previous result on EN-FR. We think it’s due to the use of a different batch size and training steps.
>
> Re: "(4c) There are no BLEU scores for the LSTM setting. Note that comment (4b) and (4c) are also pointed by anonymous readers."
>
> We reported the BLEU scores for LSTM as well. All the models are tested for both token and character-level translation tasks.
>
> Re: "(4d) It is really strange for me to “trained on COCO and tested on Flickr” (See the title of Table 4). It is not a common practice in image captioning literature. Even if in (Soricut et al., 2018), the authors report the results of training of COCO and test on COCO (the Table 5). Therefore, the results are not convincing. You should train on COCO and test on COCO too."
>
> We added COCO40 test results (from the official COCO challenge website) in the revised paper (see Table 5). Area attention significantly improved the original model on both CIDEr and ROUGE-L metrics.
>
> Re: "What if we use different area size? I do not find the study in this paper."
>
> This is a great question. We reported the effect of different area sizes for image captioning tasks, e.g., 2x2 versus 3x3. We explored which layers in Transformer can benefit from area attention and found area attention helps more when it’s used in lower layers. We speculate that this is because each position is well equipped with contextual information due to self attention in latent layers and area attention might not be able to help much. In contrast, the lower layers can benefit a lot from area attention. We also found a large area size does not necessarily improve accuracy. It is worth tuning the max area size as a hyper parameter to achieve even better results for specific problems.

---

> > ### Comment · AnonReviewer3 · 2018-12-14
> > **Response to the rebuttal**
> >
> >
> > Thanks a lot for your kind response and most of my questions are answered. This version is more solid than the initial submission. However, I still have several concerns for this work:
> >
> > [Motivation]
> > I am not fully convinced by the current motivation. (1) Without area attention, what is the phenomenon/problem of the current neural machine translation and image captioning? How many cases are suffering from the absence of area attention? Can these problems be solved by carefully tuning the network? (2) After introducing area attention, how the problems are solved? The area attention weights should be visualized. Currently, I do not find the visualization, statistics or examples in this paper.
> >
> >
> > [Significance]
> > The improvement on NMT is too minor under the transformer_big setting. In Table 1, the BLEU score of En-De of regular attention is 29.43, and area attention improves it to 29.68; on En-Fr, the improvement is 0.18, which is not significant. Since the ``area attention’’ model needs more parameters, I am not sure whether the improvement is brought by more parameters. Besides, the improvement of en-fr under transformer_base setting is also small (39.10 v.s. 39.28). Therefore, we cannot conclude that area attention really works for NMT.

---

> > > ### Author Response · Authors · 2018-12-17
> > > **Thanks for your further comments**
> > >
> > > Re: Motivation
> > > We agree that attention visualization would be a good way to show, which we should add to the paper. We feel the need to support areas or ranges has been well demonstrated in a collection of previous works that are suggested by Reviewer1 and discussed by us in the Related Work section. Area attention we proposed here provides a simple and effective solution for enabling areas. We will add concrete examples to the paper.
> > >
> > > Re: Significance
> > > On one hand, we agree that Transformer_big is a strong baseline and our gain is not always significant. On the other hand, we feel area attention provides a simple solution that can be easily applied to many tasks and models that lead to better accuracy. We want to point out that while the baseline models (Transformer_Base & Big) were extensively tuned for the NMT tasks, we did not particularly tune model conditions for area attention. We simply used all the hyperparameters that were tuned for the baselines for area attention experiments. In our recent experiments, by tuning on the maximum area size, we found area attention achieved even better accuracy, e.g., 29.74 for En-De and 41.5 for En-Fr, which enable a larger gain over the baselines. There are other hyperparameters to tune such as attention dropout ratio, which can be crucial for area attention to realize its potential. Please note that the basic form of area attention (Eq.3) requires no additional parameters and the feature combination version of area attention (Eq.5-9) uses very few additional parameters: less than 0.03% for Transformer Big.

---

### Official Review · AnonReviewer1 · 2018-11-02
**Some important related studies are missing.**

**Rating:** 6
**Confidence:** 4

**Review:**


I have several concerns about this paper.

[originality]
Some important related studies are missing.

# Related studies about the perspective of “area”.
The consecutive position in sequence is often referred to as “span” in NLP filed, which is identical to what the authors call “area” in this paper.
Then, the idea of utilizing spans currently becomes a very popular in NLP field. We can find several papers,
e.g.,
Wenhui Wang, Baobao Chang, “Graph-based dependency parsing with bidirectional lstm”, ACL-2016.
Mitchell Stern, Jacob Andreas, Dan Klein, “A Minimal Span-Based Neural Constituency Parser”, ACL-2017.
Kenton Lee, Luheng He, Mike Lewis, Luke Zettlemoyer, “End-to-end Neural Coreference Resolution”, EMNLP-2017.
Nikita Kitaev, Dan Klein, “Constituency Parsing with a Self-Attentive Encoder”, ACL-2018.

Similarly, there are several related studies in image processing field,
e,g.,
Marco Pedersoli, Thomas Lucas, Cordelia Schmid, Jakob Verbeek, “Areas of Attention for image captioning”, ICCV-2017
Quanzeng You, Hailin Jin, Zhaowen Wang, Chen Fang, and Jiebo Luo, “Image Captioning with Semantic Attention”, CVPR-2016.

# Related studies about the perspective of “structured attention”.
Several papers about structured attention have already been proposed,
e.g.,
Yoon Kim, Carl Denton, Luong Hoang, Alexander M. Rush. “Structured Attention Networks”, ICLR-2017.
Vlad Niculae, Mathieu Blondel. “A Regularized Framework for Sparse and Structured Neural Attention”, NIPS-2017.


I think the authors should explain the relations between their method and the methods proposed in the above listed papers.


[significance]
# Concern about experimental settings
The experimental setting for NMT looks unnormal in the community.
Currently, most of papers use sentences split in subword units rather than character units. I cannot find a reason to select the character units. I think the authors should report the effectiveness of the proposed method on the widely-used settings.


# computational cost
The authors should report the actual calculation speed by comparing with the baseline method and the proposed method.
In Sec. 2.2, the authors provided the computational cost.
I feel that the cost of O(|M|A) is still enough large and that can unacceptably damage the actual calculation speed of the proposed method.



Overall, the proposed method itself seems to be novel and interesting.
However, in my opinion, writing and organization of this paper should be much improved as a conference paper. I feel like the current status of this paper is still ongoing to write.
Thus, it is a bit hard for me to strongly recommend this paper to be accepted.

---

> ### Author Response · Authors · 2018-11-26
> **Responses to Reviewer1's comments**
>
> Thank you so much for the thorough feedback that really strengthens the paper. We have conducted additional experiments and revised the paper to address these points. We here respond each point in your review.
>
> Re: originality
> Thanks for bringing up a great collection of previous works, which helped us better position our work in the literature. We have added the Related Work section to clarify the relationship of area attention with each previous work you suggested.
>
> Re: Concern about experimental settings
> We added token-level translation experiments as requested. The results are summarized in Table 1 & Table 2. Area attention consistently outperformed regular attention on token-level translation on all the conditions as well. There is particularly a significant accuracy gain on EN-DE tasks.
>
> Re: computational cost
> We reported the actual calculation speed of area attention in comparison to regular attention for two major model configurations (Transformer Base and Big) in section 4.1.1. Briefly, for Transformer Base model, on 8 NVIDIA P100 GPUs, each training step took 0.4 seconds for Regular Attention, 0.5 seconds for the basic form of Area Attention (Eq.3 & Eq.4), 0.8 seconds for Area Attention using multiple features (Eq.9 & Eq.4).
>
> We have made a number of other improvements to the paper. Please see the summary of changes and the revised paper for details.

---

> > ### Comment · AnonReviewer1 · 2018-12-15
> > **Thank you for your response**
> >
> > Authors properly addressed all of my questions.
> > I agree that the paper was substantially improved.
> > I understand the authors' effort.
> > However, unfortunately, token-level translation experiments did not much support the usefulness of the area attention.
> > In conclusion, it is a bit hard to "strongly" recommend this paper to be accepted in my opinion.

---

> > > ### Author Response · Authors · 2018-12-17
> > > **Thanks for your further comments**
> > >
> > > Thank you for your encouraging remarks. We agree for some conditions our accuracy gain is not as significant. However, the accuracy improvement has been very consistent across most model conditions when area attention is used. We currently simply used the hyperparameters tuned for the baseline models. We believe some tuning of hyperparameters for area attention can result in better accuracy. In our recent experiments, area attention achieved 29.74 for En-De and 41.5 for En-Fr (a larger margin than previously reported in the revision) by simply tuning on the maximum area size. Tuning on other hyperparameters such as attention dropout ratio can further realize its potential.

---

### Public Comment · (anonymous) · 2018-09-28
**Some concerns about this paper**

Hi, this idea is interesting, though need some explanations, can you explain some my concerns regarding the motivation and experiments?

1. How are you sure that "Although softmax (Equation 1) assigns non-zero probablities to every item in the memory, it quickly converges to the most probable item due to the exponential function used for calculating probabilities" ? Can you prove this claim mathematically ? If not, is there any research out there already confirm this? if so, please cite.
I am not going to prove you wrong, but I have seen many examples that softmax doesn't converge into 1 single item. Look at some ImageNet classification papers (resnet, densenet....), the real life softmax scores distribute a lot more evenly. I guess it depends on the data, not by the convergence of softmax function.

2. eqn 6 indicates ei has dimension 1xD, miu_i, sigma_i possibly also 1xD. So the term behind the Relu function will be also 1xD. Then the whole eqn 7 : W_d x relu(µi + σi + ei; θ) is a matrix dot product of DxD and 1xD, which is dimensionally incompatible?
Correct me if i'm wrong.

3. equation kri = Wdφ(µi + σi + ei; θ) looks weird. It is unusual to sum up the mean and variance together. variance is 2-degree term, should it be added to the mean (1-degree term)? Please justify.
To me, it makes more sense to sum the mean the standard deviation rather than the variance.

4. (Vaswani et al., 2017) experimented translation with BPE tokens, which already achieved more than this paper did with character-level experiments. What is the motivation to use character-level but not (at least) BPE or word-level translation?  Why not do BPE experiment to compare with Vaswani et al.

5. What happen if the elements in a particular area got reordered? Will the result after the attention be different or the same?

Thank you,

---

> ### Public Comment · (anonymous) · 2018-09-29
> **the notation for variance is usually \sigma^2, not \sigma**
>
> I think in Eq. (5), it is more suitable by using \sigma^2, not \sigma, to denote the variance.

---

> > ### Author Response · Authors · 2018-10-02
> > **Good suggestion**
> >
> > Yes! Thanks for catching this. We will fix it in the revision.

---

> ### Author Response · Authors · 2018-10-02
> **Clarifications**
>
> Thank you for bringing up these questions. We briefly clarify them here and will address them further in the revision.
>
> Re: Question #1
> We agree that the peakiness of softmax depends on data and tasks. Our statement about softmax convergences is mostly empirical, from both previous work and our own observation. For example, the early attention work (https://arxiv.org/pdf/1409.0473.pdf) by Bahdanau, Cho & Bengio showed that it is often a single or very few items that are receiving most attention probability (see Figure 3 in the paper). Similar phenomena were reported in the Transformer work (https://arxiv.org/pdf/1706.03762.pdf) (In Figure 3, each row that uses color density to represent attention distribution is mostly white). In our own experiments, we found the entropy of the attention probability distribution tends to decrease rapidly, which indicates that the attention probability distribution is towards more deterministic rather than evenly distributed as the training proceeds. That said, we will clarify that our statement is empirical and cite the literature.
>
> Re: Question #2
> Thanks for pointing out the issue. There should be a transpose over relu(). Alternatively it should be relu(µi + σi + ei; θ) x W_d. We will correct this in the revision.
>
> Re: Question #3
> We explored both standard deviation and variance in the early experiments from which we did not see a noticeable difference. In φ, the sum of the three is projected before ReLU. That said, we agree it makes more sense to use standard deviation rather than variance, and we will run full experiments on standard deviation and report back.
>
> Re: Question #4
> Our motivation to study area attention on character-level instead of token-level comes from the fact that there are simply more areas to attend to on characters. Since many sentences only have a few tokens, it is intuitively less clear why attending to areas should help in that case. Having said that, area attention is a general framework that is applicable in all cases (e.g., image captioning as we presented), in the worst case performing similarly to plain attention. We will run token-level experiments as well and report back.
>
> Re: Question #5
> The overall attention distribution will be different, even though the representation for that specific area is the same. This is because area attention allows overlapping areas. The change in the order of items will cause these items to be picked up by different areas. For example, assume there is a sequence with six items: A, B, C, D, E, and F. Say Area 1 contains A, B and C; Area 2 contains C and D; and Area 3 contains D, E and F. Reordering C and D in Area 2 will not change Area 2’s representation. However, the reordering will leave Area 1 with A, B and D, and Area 3 with C, E and F.

---

> > ### Public Comment · (anonymous) · 2018-10-19
> > **Not convinced**
> >
> > #1
> > This is not convincing. The papers you mentioned only shows a figure of attention probability of an example in the dataset. Those were just for demonstration with a single example, not an overall observation of the entire dataset, so we cannot infer a single figure as a universal truth. I think they did not claim or proved that softmax score converge to a particular item in their papers either. They also didn't show any statistics supporting this claim.
> >
> > In your experiments, there is no statistic data supporting "entropy of the attention probability distribution tends to decrease rapidly".
> >
> > Anyway, there is nothing bad for attention softmax score to concentrate on a particular item, and that's what attention with softmax is used for. Attention acts as a word to word alignments, it should, inversely, attend on a particular item rather than spreading out equally.
> >
> > #4 transformer is simply not suitable for character-level (24.65 BLEU vs 27.3 in the original paper using BPE). This is obvious because it is harder to attend on sequence of hundreds to thousands characters than to attend on less than 100 words. Your solution makes the character-level problem easier and it makes sense. But there should not be a problem to begin with because they use token-level (BPE) translation. A possible reason (other than BLEU) to prefer character-level is to construct UNK words or rare words, but BPE already did that and there is no analysis on this in the paper anyway.
> >
> > Hope the results on token-level translation are better than original transformer, otherwise, it seems the paper just create a problem for the proposed model to work.
> >
> > #5 agree that overall attention distribution will be different. But invariance within a region is important. For instance, what is the difference between "army" and "mary"? In such case, how can overlapping the characters with nearby words might be helpful to differentiate "army" and "mary"?
> >
> > # 6 Why the paper didn't report LSTM on BLEU but perplexity? Perplexity is usually not a good indication of quality of translation rather than BLEU, the transformer paper also said that they sacrifice perplexity for better BLEU. I'm not 100% sure, but improvement in order of 0.0001 sounds not significant though.
> > Can you report BLEU of LSTM?
> >
> > #7 This can be good for image captioning though. But I guess you miss Image Transformer paper (https://arxiv.org/abs/1802.05751). That is quite similar to this paper, should cite it.
> >
> > #8 can you release the code?
> >
> > Overall:
> > + The motivation and problem (translation) is not convincing, proved or supported with statistical data to begin with. It is not shown that such characteristics of softmax (low entropy on convergence) is a problem for attention mechanism either.
> > + For translation, unless token-level experiments work, purposefully using character-level task seems just to create a problem in which area attention has the advantage over normal attention.
> > + Image caption task seems promising though. Perhaps it is more suitable and convincing to use this for vision than NLP.

---

> > > ### Author Response · Authors · 2018-10-19
> > > **Response on motivation, token-level performance and other points**
> > >
> > > Thanks again for your further comments. Please see our responses here.
> > >
> > > Re: #1
> > > We did not intend to argue that focusing on a single item is always bad. Rather, area attention is simply to give the model the option to attend to a range of items that are structurally adjacent when needed. Please notice that area attention subsumes single-item attention rather than excluding it. For example, area attention with max_area_size=5 includes all the areas with size 1-5, where area_size=1 is effectively single-item attention.
> > >
> > > Again, we will clarify our point regarding attention convergence in the revision. To answer your question, we did observe that, for example, mean entropy started with 5.0 at the initial stage of training for some layers and then dropped to less than 1.8 within the first 20K iterations. That said, we want to clarify that attention convergence is not the issue we are tackling, which is what it should be doing as you pointed out. Rather, our main motivation with area attention is to give the model more options when attending to items.
> > >
> > > Re: #4
> > > We have run experiments on token-level translation, and found area attention outperformed the transformer baselines in most conditions as well. In particular, for the Transformer (Base Model) that you are questioning, area attention achieved BLEU scores: 28.17 (ende) 39.22 (enfr). All these BLEU scores are higher than what were previously reported in the Transformer paper (ende: 27.3 and enfr: 38.1). The performance gain is simply achieved by just using the parameter-free version of Area Attention. We will add a section and a table in the revision to report the token-level performance of area attention with Transformer.
> > >
> > > Re: #5
> > > Your example is assuming each item (character) is not carrying its positional information. For Transformer, this is not the case because each item encodes its position in the sequence (see Sec 3.5: Positional Encoding in the Transformer paper https://arxiv.org/pdf/1706.03762.pdf). For LSTM, this is also less of an issue because attention is applied to the output of an LSTM layer which already captures order information to a certain extent.
> > >
> > > That said, we see your point, and we could have used a sequence model to encode each area to directly capture the order of items in the area. However, one important advantage of area attention is that it is fast and takes constant time to compute (key, value) for each area due to the use of summed area table, and its basic form (Eq.3 & 4) is parameter free.
> > >
> > > Re: #6
> > > We reported perplexity to see the relative trend when area attention is in use, which showed that area attention often improved over regular attention in LSTM. That said, we will report back the BLEU scores for LSTM.
> > >
> > > Re: #7
> > > We will cite the Image Transformer paper. It uses regular multi-head attention and solves a different task on conditional image generation.
> > >
> > > Re: #8
> > > Yes. We will release the code as we implemented area attention directly based on the open source Tensor2Tensor library (https://github.com/tensorflow/tensor2tensor), where benchmark Transformer models and tasks are implemented, which guarantees a solid comparison with regular attention in Transformer.
> > >
> > > Overall:
> > > We clarify that our motivation with area attention is to give a model more options when attending to items. Area attention subsumes regular-attention rather than excluding it as explained earlier. Even the parameter-free version of area attention has outperformed regular attention on both character-level and token-level translation tasks, and image captioning tasks. We will add new experimental results to the revision.

---

> > > > ### Public Comment · (anonymous) · 2018-10-21
> > > > **Wonderful**
> > > >
> > > > Thank you for the prompt response.
> > > >
> > > > Now the motivation is clear. "Giving the model more options to attend to a range of items that are structurally adjacent when needed" is a nicer motivation rather than arguing the softmax convergence and focusing on a single item is bad.
> > > > You may want to make this point clear in the revision because the initial writing makes me feel the motivation is to tackle softmax convergence and the reviewers can possibly interpret like this too.
> > > >
> > > > Now token-level translation also works. Good jobs.

---

> > > > > ### Author Response · Authors · 2018-10-23
> > > > > **Thanks!**
> > > > >
> > > > > Yes. We will make this point clear in the revision. Thanks for these suggestions that strengthen the contribution of our paper.

---

### Public Comment · (anonymous) · 2018-10-02
**Some implementation or reproduction problem about this paper**

Interesting work, but the algorithm seems to be ambiguous. Hope you can release the code to verity the details.

---

> ### Author Response · Authors · 2018-10-02
> **Code**
>
> Thank you for your interest in reading the work. We will make the pseudo code more readable and release the source code that is written in TensorFlow. Our experiments were conducted based on the original Transformer implementation released in Tensor2Tensor (https://github.com/tensorflow/tensor2tensor).

---

### Public Comment · (anonymous) · 2018-10-03
**Some related work**

I find this is an interesting approach to attention that could be broadly applicable.
I have been interested in those approaches for some time and am curious to see how it devellops.

Here is an reference that is relevant:  https://arxiv.org/pdf/1612.01033.pdf
(Areas of Attention for image captioning)

Cheers

---

> ### Author Response · Authors · 2018-10-08
> **Will cite & discuss the related work**
>
> Thank you for bringing up this previous work that is indeed relevant. The paper focused on image captioning and proposed two nice methods for attending to object regions on images, where both use a special network to infer regions to attend. In contrast, our method examines all possible areas with summed area table for fast computation. The basic form of area attention we proposed is parameter free. We also intend to propose area attention as a general mechanism beyond captioning tasks. We will cite and discuss the paper in the revision.

---

### Author Response · Authors · 2018-11-26
**Summary of Changes**

We thank the anonymous reviewers and readers for their thoughtful feedback and encouraging remarks. We have substantially improved the paper by making the following changes in the revision.

1. Motivation & Goals
Clarified the purpose of area attention throughout the paper. Area attention allows a model to attend to information with learned, varying granularity and levels of aggregation, which is in contrast to existing attention mechanisms focus on items with predetermined fixed granularity. Please read the revision for details.

2. Related Work
Added a Related Work section to clarify the relationship of area attention with each previous work suggested by Reviewer1 and readers.

3. Presentation
Increased the clarity of the writing throughout the paper, particularly improved the Pseudo code.

4. Experiments
* Token-Level Translation
Added a section for token-level translation experiments as requested by the reviewers and readers. The results are summarized in Table 1 & 2. Area attention consistently outperformed regular attention on token-level translation across all the conditions. In particular, it improved upon the state-of-art result on EN-DE with a significant margin.

* Actual Cost
Reported the actual calculation speed of area attention in comparison to regular attention for two major model configurations (Transformer Base and Big) in section 4.1.1.

* Comparing Area Attention Keys
Added a comparison of the two forms of area attention keys: the parameter free version (Eq.3) and the feature combination version (Eq.5-9) on all the token-level translation tasks (see Table 1 & 2).

* BLEU for LSTM
Reported BLEU scores for LSTM translation tasks as well (see Table 2 & 4).

* Transformer Big
Added experiments with Transformer Big where area attention has also shown consistent improvements.

* Tests on COCO
Added COCO40 official tests for in-domain image captioning (see Table 5). Area attention outperformed the benchmark model with a significant margin.

---

### Meta-Review · Area_Chair1 · 2018-12-18
**reject**

**Confidence:** 3
**Recommendation:** Reject

**Metareview:**

although the idea is a straightforward extension of the usual (flat) attention mechanism (which is positive), it does show some improvement in a series of experiments done in this submission. the reviewers however found the experimental results to be rather weak and believe that there may be other problems in which the proposed attention mechanism could be better utilized, despite the authors' effort at improving the result further during the rebuttal period. this may be due to a less-than-desirable form the initial submission was in, and when the new version with perhaps a new set of more convincing experiments is reviewed elsewhere, it may be received with a more positive attitude from the reviewers.